# Archean phosphorus liberation induced by iron redox geochemistry

Barry Herschy[1], Sae Jung Chang[2,3], Ruth Blake[3], Aivo Lepland[4], Heather Abbott-Lyon[5], Jacqueline Sampson[1], Zachary Atlas[1], Terence P. Kee[6] & Matthew A. Pasek[1]

The element phosphorus (P) is central to ecosystem growth and is proposed to be a limiting nutrient for life. The Archean ocean may have been strongly phosphorus-limited due to the selective binding of phosphate to iron oxyhydroxide. Here we report a new route to solubilizing phosphorus in the ancient oceans: reduction of phosphate to phosphite by iron(II) at low (<200 °C) diagenetic temperatures. Reduction of phosphate to phosphite was likely widespread in the Archean, as the reaction occurs rapidly and is demonstrated from thermochemical modeling, experimental analogs, and detection of phosphite in early Archean rocks. We further demonstrate that the higher solubility of phosphite compared to phosphate results in the liberation of phosphorus from ferruginous sediments. This phosphite is relatively stable after its formation, allowing its accumulation in the early oceans. As such, phosphorus, not as phosphate but as phosphite, could have been a major nutrient in early pre-oxygenated oceans.

[1] School of Geosciences, University of South Florida, 4202 E Fowler Ave NES 204, Tampa, FL 33620, USA. [2] Seoul Center, Korea Basic Science Institute, Seoul 02841, Republic of Korea. [3] Department of Geology and Geophysics, Yale University, New Haven, CT 06520, USA. [4] Geological Survey of Norway, 7491 Trondheim, Norway. [5] Department of Chemistry, Kennesaw State University, Kennesaw, GA 30144, USA. [6] School of Chemistry, University of Leeds, Leeds LS2 9JT, UK. Correspondence and requests for materials should be addressed to M.A.P. (email: mpasek@usf.edu)

P hosphorus limitation in the Archean eon has been proposed to have limited the fecundity of the early Earth[1–4]. Although the extent of removal by iron oxyhydroxide (IOH) has been a matter of question due to competition by silica[5, 6] and arsenic[7, 8], a phosphorus-depleted Archean ocean would have affected the development and diversification of life on the Earth. The effect of diagenesis on Archean phosphorus (P) remobilization is largely unknown, due to numerous factors such as oxygen levels, sedimentary rates, and temperature[9] that may affect the phosphate speciation and mineralization. The main pathway for remobilization of sedimentary phosphate today is through the decay of organic matter, although the early Archean microbial reduction of ferric to ferrous iron may have remobilized IOH-bound P, as ferrous phosphates are significantly more soluble in water[10]. Remobilization would have allowed the release of P into the interstitial waters, though some would have been rebound by IOH in overlying sedimentary layers. The operating hypothesis for phosphate diagenesis is that phosphate does not change oxidation state; in most geochemical studies, phosphates are considered synonymous with total phosphorus.

An alternative to phosphate is the ion phosphite ($HPO_3^{2-}$), an ion that is more soluble in the presence of divalent cation salts than phosphate. Geochemical occurrences of phosphite are sparse: it has been found in trace amounts in hydrothermal water systems of Hot Creek Gorge, Mammoth Lakes, CA[11], and within some rocks struck by lightning[12]. Mulkidjanian et al.[13] proposed that the geothermal reduction of phosphate to phosphite occurs principally by high-temperature heating in the presence of carbon and iron, within volcanic hydrothermal systems thought to be common on the early Earth. After the discovery of phosphite in geothermal waters by Pech et al.[11], reduced P was also found in modern, natural water samples[14, 15], though biological redox cycling is the likely source of phosphite in these systems. These results were geochemically surprising, as the prevalent oxidizing conditions on Earth should confine P to the pentavalent oxidation state, and P occurs principally as the mineral apatite in surface rocks[16]. In contrast, the ability to use reduced P compounds as a sole P source is widespread in microbiological systems[17, 18]. These organisms, such as the bacteria, *Desulfotignum phosphitoxidans*, can use phosphite as part of their metabolic cycle[19]. If these organisms are ubiquitous, their widespread occurrence is consistent with the presence of unrecognized sources of reduced P species.

We propose here that phosphorus may not have been ecosystem-limiting on the early Earth, because of the existence of a hitherto unrecognized phosphate reduction process extensively active on the early Earth. We demonstrate that phosphate was reduced to phosphite by ferrous iron, even at the low temperatures (<200 °C) of sedimentary diagenesis. Such conditions would have been widespread on the Archean Earth, before the oxidation of the atmosphere, known as Great Oxygenation Event (GOE) at c. 2.4 Ga[20–22]. In the nominal reaction, phosphate ($HPO_4^{2-}$) is reduced to phosphite ($HPO_3^{2-}$) by the concurrent oxidation of iron (II). Phosphite formed through reaction with iron(II) would have been extensively mobilized because of its high solubility in water, compared with phosphate, and the total P concentrations may have approached 100–1000 times those predicted from IOH binding experiments. The presence of reduced phosphorus compounds would hence not be limited to the requirements of meteoritic impact and weathering for release[23, 24], and could potentially account for the elevated levels of phosphite present in early Archean carbonates[25].

## Results

**Archean rock analysis.** Pasek et al.[25] first reported the presence of phosphite in early Archean sedimentary carbonates from the

**Table 1 Sample site designation, rock types, and phosphorus speciation**

| Sample name | Rock location and type | $HPO_3^{2-}$ ppm (P³⁺) | $PO_4^{3-}$ ppm[a] (P⁵⁺) |
|---|---|---|---|
| ISUA-AL7-2 | Isua metacarbonate | 1.1 | 10 |
| ISUA-AL-44 | Isua metacarbonate | 13 | 0.43 |
| ISUA-AL-8-1 | Isua metacarbonate | 0.85 | 22 |
| ISUA-AL-42 | Isua metachert | 21 | 10 |
| AKILIA-AK98 | Akilia quartzite-amphibole-pyroxene rock | 110 | 34 |
| ISUA-AL35-6 | Isua BIF | 10 | 32 |
| ISUA-AL-15 | Isua BIF | 27 | 32 |

[a] Note: phosphorus was extracted using acetic acid and sodium acetate. All phosphate may not have been extracted using this method, and total P content is typically around 1000 ppm (or, as $PO_4^{3-}$, about 2500 ppm)

Pilbara Craton, Australia, and attributed this phosphite to reactive, meteoritic sources of phosphorus on the early Earth. Here, we expand on the suite of early Archean rocks and report data on the samples collected from the supracrustal succession of the Isua belt and the Akilia enclave, western Greenland. The age on the Isua Supracrustal Belt is 3.7–3.8 Ga[26], whereas the age of the Akilia enclave is controversial, either c. 3.85 Ga[27, 28] or c. 3.65 Ga[29, 30]. Both Isua and Akilia rocks have experienced deformation, metasomatism, and metamorphism, that in Isua has reached amphibolite facies, and in Akilia, granulite facies[27, 31–35]. Isua samples were collected from the "low strain" domain[36] in the northwestern part of the belt and represent chemical sediments, such as banded iron formations (BIF) and metacherts, and metacarbonates that have metasomatic origin. Apatite occurring in BIF and metacherts have trace element signatures consistent with Archean seawater[37]. The studied Akilia sample is from the quartz-amphibole-pyroxene unit. The protolith of that unit, either chemical sediment or metasomatised ultramafic rock[27, 34], has been debated in the literature. The trace element characteristics of the whole-rock and apatite in the best preserved parts of the quartz-amphibole-pyroxene unit are consistent with a sedimentary origin[38].

The samples were analyzed, using High Performance Liquid Chromatography-Inductively Coupled Plasma-Mass Spectrometry (HPLC–ICP–MS), to identify P speciation within the rock extracts (Table 1, Fig. 1, Methods, Supplementary Fig. 1). The analyses show that phosphite was present in these rock samples at levels of 1–100 ppm by weight (as $HPO_3^{2-}$), or about 0.01% to a few percent of the total P. These findings show that phosphite was indeed present in Archean rocks, and that phosphite may have been common in early oceans, especially since reduction occurs rapidly, and phosphite is readily extractable (see below). Although the source of phosphite was not established in the Pilbara carbonates, it was previously hedged to be meteoritic[25], because of the high flux of meteorites to the Earth in early Archean and Hadean[23]. At present, we do not know the phase associated with this phosphite in these rocks, nor do we know if the phosphite was formed during diagenesis or metamorphism, or even if it was captured when the rocks were initially deposited.

**Thermodynamic modeling.** In addition to meteoritic sources, reduction of phosphate to phosphite by abiotic redox reactions may have provided phosphite to the Archean oceans. Though the source of phosphite in these Archean rocks cannot be specifically identified, the reduction of phosphate by Fe(II) is facile. We modeled the reduction of phosphate by common geochemical reducing agents, including hydrogen, ammonia, hydrogen sulfide,

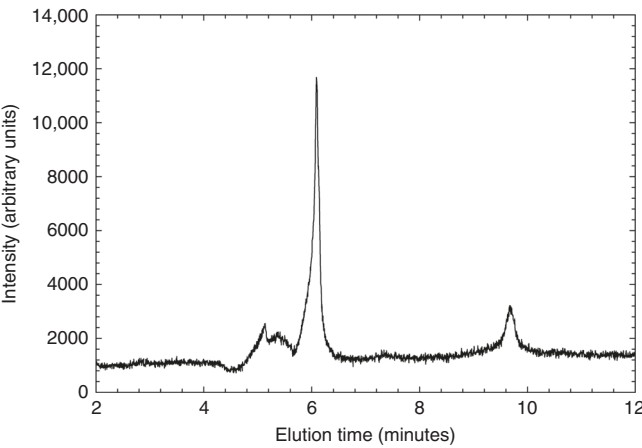

**Fig. 1** Chromatogram of phosphorus speciation of sample AKILIA-AK98. $m/z + 47$ (PO) was observed by ICP–MS. The peak at 6 min matches phosphite and the peak at 9.5 min matches the retention time of phosphate (Methods)

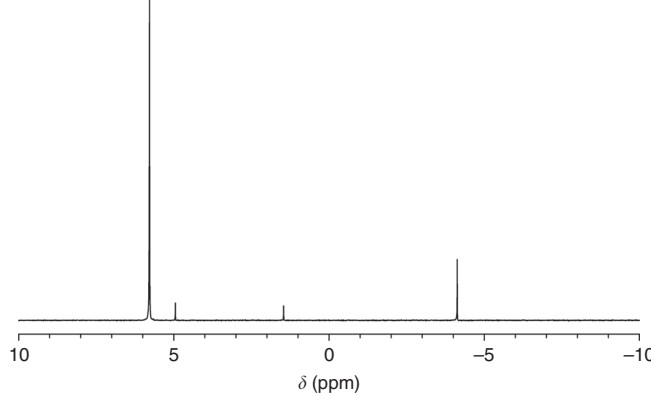

**Fig. 2** NMR spectrum of heated $Fe^{2+}$ and phosphate solution. Proton-coupled $^{31}P$ NMR of a solution of $FeCl_2$ and $Na_2HPO_4$ heated to dryness (180 °C), under flowing $N_2$. All compounds are references to an external standard of 85% $H_3PO_4$ (0 ppm, as a frequency spectrum referenced to $H_3PO_4$ at 161.9 MHz). Phosphate is at 5.8 ppm, and pyrophosphate ($HP_2O_7^{3-}$) is at −4.2 ppm. The small doublet at 4.95 and 1.46 ppm corresponds to phosphite (4% of total area), identified by its large H–P J-coupling constant of 565 Hz[23]

and ferrous iron, among others (see Methods, Supplementary Figs. 2, 3), with the result showing that only a few inorganic reagents are capable of reducing phosphate. Reduction of phosphate by ferrous iron (as FeO) occurs up to a few percent of the total phosphorus at relatively low temperature (<200 °C), whereas other reductants either do not reduce phosphate to any significant extent, or do so at high temperature (>500 °C). However, given that ferrous iron was likely stable on the Earth's surface before GOE, it is quite likely that reduction of phosphate by low-temperature oxidation of iron is the source of phosphite within the Archean rocks.

**Experimental phosphate reduction**. Motivated by the feasibility of reduction from thermodynamic modeling, phosphate reduction by ferrous iron is demonstrated by laboratory simulation. Phosphate, when heated with iron (II) to 160–200 °C under a dinitrogen atmosphere, is reduced to phosphite at up to 4% of the total P speciation (Fig. 2, see Methods). These experiments confirm that moderate heating of phosphate in simulated sedimentary diagenetic or low temperature hydrothermal conditions[26] in the presence of ferrous iron, can lead to production of phosphite, after only a few days of heating. Under higher temperatures (>200 °C), the models described above predict that more phosphate will be reduced, but experimentally we find that formation of the phosphate dimer (pyrophosphate, $P_2O_7^{4-}$) competes with reduction. Since this reaction releases $H_2O$, it is possible that under higher pressures reduction is favored, instead of $H_2O(g)$ loss. Thus, the phosphite present within the Archean samples is most consistent with the heating of phosphate along with the oxidation of iron (II), and required only relatively low temperatures to proceed, though higher temperatures may have also led to reduction.

**Solutional experiments and modeling**. Furthermore, ferrous and ferric phosphite salts are also more soluble than the equivalent iron phosphate salts, by a factor of $10^2$–$10^4 \times$ (see Supplementary Table 1). This increased solubility could have affected the ocean chemistry substantially, as shown by a simple thermodynamic mass-balance model that incorporates the above thermodynamic calculations with iron phosphate/phosphite solubility predictions. In this model, water flowing through ferruginous sediments, containing some iron phosphate and iron phosphite formed by reduction (Methods), preferentially extracts phosphite, flushing it out to the ocean at a concentration of $10^{-4}$ M (Fig. 3a), and

requiring relatively low amounts of water to completely extract phosphite. If hydrothermal fluid fluxes[39] were active at present day rates or higher on the early Earth, the ocean could have reached iron-phosphite saturation with $10^{-4}$ M phosphite, globally, in less than one million years, assuming minimal phosphite oxidation (Fig. 3b).

**Discussion**
In such a scenario, the early Earth oceans were enriched with phosphorus as phosphite. The total phosphorus content of the early oceans could have been up to 1000× greater than if controlled by IOH-binding, as previously predicted. Given that most microbial systems primarily use phosphorus as phosphate, microbes would either have had to have waited for the (slow) oxidation of phosphite to phosphate, or would have had the genes necessary for phosphite oxidation, a similar scenario as modern microbes. Given that there is a widespread distribution of phosphite-utilizing genes in the modern microbiome[17], and that microbial strains bearing these genes diverged billions of years ago (Methods), it is possible that the ability to use phosphite as a sole P nutrient is an artifact of ancient ocean chemistry, though horizontal gene transfer may be a more likely explanation. In either case, phosphorus may not have been as critical a limiting nutrient, relative to other key elements.

This model is predicated on two hypotheses. The first is that iron and phosphorus co-occurred within the sediments in the Archean. The thermodynamic calculations here modeled this behavior, assuming P was initially in vivianite (with parallels to modern lakes)[40]; however, calcium phosphate minerals may also be susceptible to reduction to produce phosphite[12]. Iron and phosphate, even if the latter occurs in calcium phosphates, were likely closely associated in many Archean sediments, and have already been co-located in hydrothermal plumes off the southern East Pacific Rise[41], with later diagenesis perhaps altering these minerals to release phosphite.

Secondly, the oxidation of phosphite is presumed to be slow enough to allow high concentrations to accumulate. Oxidation of phosphite has both an abiotic and a biologic component. As evidence for the presumed stability of phosphite, abiotic oxidation experiments demonstrate low susceptibility for the calcium phosphite salt toward oxidation[25]. In addition, the oxidation of

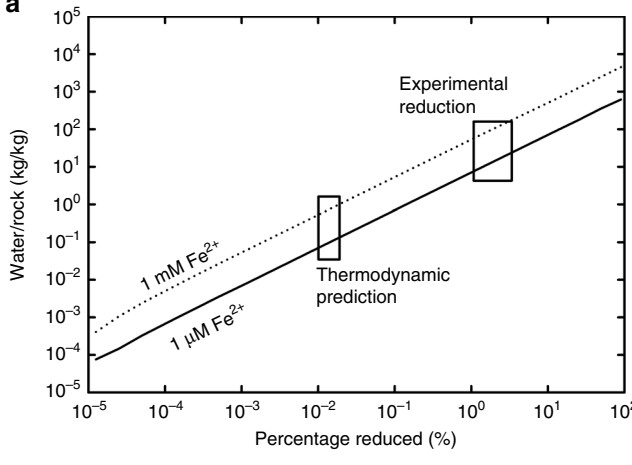

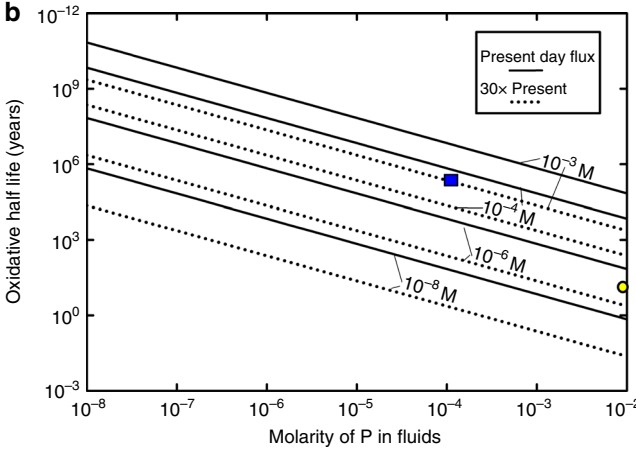

**Fig. 3** Model results of the extraction of phosphite and its predicted steady-state abundance in an anoxic ocean. **a** Water-rock ratios required to completely extract phosphite from ferruginous sediments (0.3% P, $Fe^{2+}/Fe^{3+}$ of 1/1). These calculations assume a total P content of 0.3% by weight, and that the P is in vivianite [$Fe_3(PO_4)_2 \cdot 8H_2O$] or ferric phosphite—$Fe_2(HPO_3)_3$. Then, using the solubility data determined experimentally (ED), the quantity of water (pH 7.2) needed to completely dissolve phosphite was determined using the thermodynamic modeling program HSC Chemistry (Methods). Two scenarios were investigated: one where the water in contact with the rock had a starting $Fe^{2+}$ concentration of $10^{-6}$ M, and other, with a concentration of $10^{-3}$ M. The amount of phosphite predicted to form from thermodynamics and shown to form from experimental reduction, also imply that lower quantities of phosphite produced at low temperatures (<160 °C) will also be easily extracted at low water-rock ratios. **b** Steady-state concentration of phosphite predicted within the early oceans, as a function of molarity of phosphite dissolved from rock after diagenesis, oxidative half-life of phosphite for modern day hydrothermal fluid fluxes, and the higher values that might be expected in the Archean[28]. The blue square corresponds to the predicted phosphite flux from Fig. 3a, with the measured oxidative half-life of $Fe(H_2PO_3)_2$ under low $O_2$ conditions (Methods). The yellow circle corresponds to the current measurements of half-life oxidation by biology[18] with the concentration of phosphite found within the ocean[15], which implies a very high flux from sediments or hydrothermal sources, or more likely indicating that oceanic phosphite is not currently steady state, with respect to abiotic sources. This biotic oxidation rate is likely overestimated (Methods)

ferrous phosphite is demonstrated to occur at temperatures higher than 300 °C in air, suggesting that at lower partial pressures of oxygen (~$10^{-5}$× present atmospheric level), little oxidation will occur under typical geologic conditions (phosphite oxidation half-life ~$6 \times 10^5$ years, see Supplementary Fig. 4). The

abiotic oxidation of dissolved phosphite proceeds most effectively by a radical reaction[16, 42], and oxidation of the phosphite solutions under air occurs slowly (half-life in air ~1000 years, see Methods, Supplementary Tables 2, 3), arguing that oxidation in early low $O_2$ environments should have occurred very slowly.

Biologic oxidation may have been important in the Archean, although calculations of enzymatic oxidation rates suggest that the rate of biologic oxidation is close to the predicted abiotic oxidation rate (see Supplementary Fig. 5, Methods). In addition, biologic oxidation of the reduced P is driven by a few enzymes, several of which require $O_2$ for redox[17]. This oxidation would have necessarily occurred over geologic timescales, possibly after the end of the Archean. If phosphite were a major P-bearing species in the Archean ocean, then its oxidation to phosphate near the time of the GOE 2.4 Ga could have been captured in the sedimentary record. The formation of the first global phosphorites occurred in the Proterozoic 1.7–2.2 Ga[37, 43, 44]. Before this period, phosphorites were rare. Whether or not phosphite oxidation may be linked to the large-scale phosphate deposition in the Proterozoic is unclear, but the timing of these events suggests the possibility, and the mass balance of P possibly present as phosphite in the ocean greatly exceeds the total reserves of phosphorites in Proterozoic rocks[43].

The diagenesis of ferrous iron and phosphate minerals to form phosphite in Archean rocks may account, at least in part, for the lower-than-expected P content associated with ferruginous sediment[3]. The loss of crystalline or adsorbed water by ferruginous sediments at temperatures around 200 °C will result in the preferential extraction of the phosphite ion as water escapes the sediment, resulting in a lower total P in the rock than without reduction. The removal of 75% of phosphorus from a rock requires about 100 L of water to flow through 1 kg of rock, if buffered by the solubility of ferrous phosphite. If instead, the fluids have lower iron concentrations because of the precipitation of other minerals, then the extraction is instead dependent on the amount and rate of reduction of phosphate, requiring even less fluid (Fig. 3). Although the investigation here did not measure phosphite within late Archean banded-iron formation rocks (e.g., ref. [3]), the processes active within the early Archean should not have varied significantly; reduction of phosphate requires only low-temperature diagenetic interaction with ferrous iron, which may have been favored during BIF deposition before the GOE[3].

## Methods

**Samples**. Samples were collected from Southwest Greenland during field campaigns in the previous work in the Isua Supercrustal Belt and Akilia[37, 45, 46].

**Phosphate to phosphite reduction experiments**. Dibasic sodium phosphate ($Na_2HPO_4$, 1 mM) and ferrous chloride ($FeCl_2 \cdot 4H_2O$, 3 mM) was dissolved in deionized water (10 mL) and stirred for 10 min, while adjusting to pH 4. This was reduced to dryness on a rotary evaporator, and the residues were heated under a constant nitrogen flow at 180 °C over a sand bath for 72 h. The heated residue was washed with deionized water (10 mL), filtered through 0.45 μm syringe filter before being again reduced to dryness and dissolved in deuterium oxide (1 mL) for $^{31}P$ {$^1H$} Nuclear Magnetic Resonance (NMR) analysis (Varian 400 MHz NMR at USF).

Analysis of the samples showed the presence of phosphate (singlet, 80%, 5.5 ppm), pyrophosphate (singlet, 15%, −5.5 ppm), and phosphite (doublet, 5%, 3.2 ppm). Previous studies on phosphate heating have shown the effective production of pyrophosphate at temperatures exceeding 180 °C, but never the presence of phosphite from this process[47]. To confirm that ferrous iron ($Fe^{2+}$) was responsible for the reduction of phosphate, control experiments were conducted using metallic iron ($Fe^0$) and ferric iron ($Fe^{3+}$) in place of ferrous iron, also one experiment with no iron present. Both the iron experiments and the phosphate only experiment showed the presence of only $P^{5+}$, confirming the role of ferrous iron in the reduction process. We decided to test this further and were also able to reduce phosphite ($P^{3+}$) to hypophosphite ($P^{1+}$) through the same process.

We note that the amount of phosphite obtained in our experiments was higher than expected from the in silico modeling below; this is thought because of the

model predictions using solution chemistry, while our experimental setup was non-solutional.

**Phosphite salt solubility**. Mg, Fe, Ca, and Al salts of phosphite were synthesized by the slow addition of a sodium phosphite solution (0.1 M) to a solution of metal chloride (in this case, $MgCl_2$, $FeCl_2$, $FeCl_3$, $CaCl_2$, and $AlCl_3$). In each case, the resulting precipitate was collected, and dried under vacuum. The $Fe^{2+}$ salt was stored in a glove box under $N_2$. A 0.01 g sample of each powder was then placed in 10 mL of 18.2 MΩ doubly deionized water and stirred for one day. The solutions were then analyzed by ICP–OES (Perkin Elmer Optima 2000DV at USF) to determine the metal ion concentration and phosphorus concentration, and was calibrated with standards of each ion of interest, with concentrations between $10^{-5}$ and $10^{-3}$ M. Results of each dissolution are shown as Supplementary Table 1, and compared with similar phosphate minerals.

**HPLC–ICP–MS methods**. A Perkin Elmer S200 High Performance Liquid Chromatograph, capable of coupling to a Perkin Elmer Elan DRC II inductively coupled plasma mass spectrometer (HPLC–ICP–MS at USF), was used for this study. For each sample, a 0.1 g sample of powdered rock was extracted with a 1 mL solution of 0.5 M acetic acid/sodium acetate buffer. General methods were modified from those used for P speciation with ion chromatography (IC) and IC electro-spray mass spectrometry[48–50], and optimized for HPLC–ICP–MS. A Dionex IonPac® AS17C chromatographic column, with an AG17 Guard Column fitted on the HPLC. All standards samples and blanks were mixed or prepared using 18 MΩ water or doubly distilled 18 MΩ water, distilled acids, and ACS grade or better solid reagents. Calibration and single species standards were mixed just before analysis.

Initial trials eluted 50 μL of sample with either 35 mM KOH or NaOH (recommended eluent for the AS17C column) as a mobile phase, but NaOH achieved poor separation under all conditions tried, and hence was abandoned. As liquids with high concentrations of total dissolved solids are known to deposit and coat the ICP–MS cones and lens, attempts to ascertain the lowest molar concentration that could be used to adequately separate phosphorus species were taken. We applied multiple gradients from 0% KOH, 100% water to 100% 35 mM KOH in various combinations. Best results were found using linear gradients, using a starting concentration of KOH of 3.5 mM for the first 2 min and then ramping up to 35 mM over the course of 10 min, and then remaining at 35 mM for the last 10 min at a 1.0 mL/min flow rate. Peak pressures in the HPLC did not exceed 3000 PSI during the course of the run. Preferred analytical conditions on the HPLC are provided in the previous work[14]. The ICP–MS was run at 1300 W RF power to effectively ionize phosphorus, which has a high first ionization potential. Nebulizer flow and lens voltage was adjusted to optimize for maximal signal intensity on phosphorus.

The Perkin Elmer Elan II DRC series is equipped with a reaction cell situated in line to the detector used to eliminate the interferences, with the aid of a reaction gas to either combine the species of interest into a molecular species at a different mass or by reaction with the interfering species. Following the methods used for analysis of arsenic (As with $m/z = 75$ is converted to AsO with $m/z = 91$ to avoid molecular interferences)[14], we charged the reaction cell with oxygen to convert $^{31}P$ to P–O ($m/z = 47$). This was done to reduce/avoid the interference from $^{15}N^{16}O$. Although the total abundance of $^{15}N$ is low, the atmospheric interface on the ICP nearly assures that a substantial signal from $^{15}N^{16}O$ will always be present. Therefore, conversion of $^{31}P$ to P–O should work to lower the background and enhance the signal to noise ratio in a similar fashion, as arsenic conversion to AsO[14].

A series of standards was prepared from commercially available hypophosphorous acid, phosphoric acid, and phosphorous acid in analytical grade 18 MΩ DI–$H_2O$ to make $1 \times 10^{-3}$ M solutions for each species, which in turn were serially diluted to make individual standards with concentrations ranging from $1 \times 10^{-9}$ M to $1 \times 10^{-4}$ M. Mixed species standards were also prepared over the same concentration range to check for adequate separation and detection, and to test the lower limits of instrument detection on the acquisition of P at $m/z = 31$ (standard mode) and P–O at $m/z = 47$ (reaction mode). Standards were stored in brown glasses or opaque HDPE containers for not more than 4 days at 4 °C, and then discarded. No notable concentration difference was found for species stored in HDPE vs. glass. Analytical trials were conducted with these general conditions with detection on the ICP–MS as either P at $m/z = 31$ or as PO at $m/z = 47$ to confirm which detection method offered the best response (Supplementary Fig. 1).

**Thermodynamics of reduction**. Thermodynamic equilibrium modeling was done using HSC Chemistry (version 7.1, Outokumpu Research Oy at USF). Reactions were solved using this program's reaction equilibrium calculator, including coupling phosphate ($HPO_4^{2-}$) reduction to phosphite ($HPO_3^{2-}$) with ammonium to nitrate, hydrogen sulfide to sulfate, methane to carbon dioxide, MnO to $MnO_2$, $Cr_2O_3$ to $CrO_3^{2-}$, $H_2$ to $H_2O$, and FeO to $Fe_2O_3$. Reactions were redox balanced with $H^+$ and $H_2O$ as needed, then solved for the reaction quotient $K$ over from 0 °C to 500 °C. The pH was assumed to be 7, when calculating for the fraction of P reduced (Supplementary Fig. 2).

The redox-dependent chemistry of phosphorus on iron was investigated to greater depth using the HSC equilibrium chemistry model, which uses the GIBBS energy solver[51] to determine equilibrium quantities and solve the reaction mass balances. This program has been used previously to understand water-rock

transitions[52], model Europa ocean chemistry[53], and determine lightning promoted rock vaporization thermodynamics[54]. Using this model, the phosphite to phosphate ratio was calculated for various iron/iron oxide redox buffers. Species considered include quartz (Q), iron metal (I), fayalite (F), wüstite (W), magnetite (M), and hematite (H). Phosphite is produced at the highest yield in those models that have iron or wüstite as starting reactants (Supplementary Fig. 3).

An Eh–pH diagram of phosphorus and iron is provided as Supplementary Figs. 6, 7 to demonstrate the predicted P and Fe behaviors. This diagram was designed using the Eh–pH module of HSC Chemistry, and was calculated at 25 °C and $10^{-8}$ M, for both total P and total Fe. Alternatively, a P-specific Eh–pH diagram can be found in previous work[16].

A simple dissolution model was constructed to determine how much water was required to reasonably dissolve all the phosphite formed by reduction in the HSC Chemistry program. The equilibrium chemistry was calculated as a function of the added solids FeO, $Fe_2O_3$ (2:1 by mole, 1:1 by Fe content) vivianite, strengite, and the two phosphite salts $FeHPO_3$ and $Fe_2(HPO_3)_3$, with dissolution governing the aqueous species by $Fe^{2+}$, $Fe^{3+}$, $FeOH^{2+}$, $FeOH^+$, $Fe(OH)_2^+$, $Fe(OH)_3^-$, Fe$(OH)_4^-$, $H_3PO_3$, $H_3PO_4$, $H_2PO_3^-$, $H_2PO_4^-$, $HPO_3^{2-}$, $HPO_4^{2-}$, $PO_4^{3-}$, $H^+$, and $OH^-$. Thermodynamic data for vivianite and strengite controlled the solubility of phosphate and iron in these systems[10]. The phosphite salt stabilities described in Supplementary Table 1 controlled the solubility of phosphite. All calculations were performed at 25 °C, and solved the volume of water needed to dissolve iron phosphite completely at varied starting phosphite to phosphate ratios. The total quantity of P was assumed to be 0.3 wt.%, as phosphate and phosphite. The pH of the fluid was set to 7.2, "buffered" in this case by $H_2S(aq)$ and $Na_2S$ in a 1:0.4 ratio. Ferrous iron was added by inclusion of $FeCl_2$ at amounts that completely dissolve to $Fe^{2+}$ concentrations of either $10^{-6}$ or $10^{-3}$ M.

A second model determined the steady-state concentration of phosphite from abiotic sources. At steady state, the amount of phosphite released from rock is expected to balance the amount of phosphite oxidized by abiotic processes, ignoring further oxidation from biological sources, summarized by:

$$\frac{\lambda F_{HT}}{M_{Ocean}} = [P^{3+}]\frac{\ln(2)}{t_{1/2}} \tag{1}$$

where $\lambda$ is the molarity of phosphite in the dissolving rock, $F_{HT}$ is the flux of hydrothermal fluid carrying this concentration of phosphite (estimated[55] as ~$10^{15}$ kg/yr for modern day), $M_{Ocean}$ is the mass of the ocean ($10^{21}$ kg), $[P^{3+}]$ is the steady-state concentration (molarity) of phosphite in the ocean, globally, and $t_{1/2}$ is the oxidative half-life of phosphite. Assumed phosphite concentrations at steady state, then yield a relationship between $\lambda$ and $t_{1/2}$, as neither have values that are well constrained at present. In this approach, the possibility of phosphite dilution or decreased yield from rock is thus factored in by altering lambda.

**Solid oxidation experiments**. $Fe(H_2PO_3)_2$ was synthesized by the acid-metal method as follows: a solution of phosphorous acid ($H_3PO_3$, 1 M) was prepared in 20 mL deionized water. The iron-metal powder (325 mesh, 3.351 g) was slowly added with stirring and was left overnight under flowing $N_2$ gas. The precipitate was collected by vacuum filtration, then dried under vacuum for 12 h before analysis.

Thermogravimetric Analysis (TGA) was performed using a TGA-50 (Shimadzu Scientific Instruments at KSU). A 9.700 mg sample of $Fe(H_2PO_3)_2 \times 2H_2O$ was heated under air in an aluminum pan with a ramp rate of 10 °C/min between room temperature and 130 °C and a ramp rate of 5 °C/min between 130 and 500 °C, with mass and sample temperature measured at each second (Supplementary Fig. 4).

**Microbial divergence**. Two bacterial species capable of utilizing phosphite are *Pseudomonas stutzeri* and *Prochlorococcus*. Both are capable of utilizing phosphite as a sole phosphorus source[56, 57]. The divergence of these two species was estimated, using the TimeTree knowledge-base[58]. These results are consistent with arguments made by previous workers[17, 59].

**Solutional oxidation experiments**. A set of experiments were designed to measure the oxidation of phosphite in aqueous solution under air. Two experiments were set up to promote oxidation, employing iron metal. Iron metal initiates oxidation of phosphite in the presence of oxygen[60] by reacting with $O_2$ to form OH radicals[61], which oxidize phosphite[42]. Two other experiments were set up in deionized water, and a final control experiment used no phosphite, only iron (to test P release from the iron reagent).

The conditions of experiments are given in Supplementary Table 2. A total 30 mM of phosphite sodium phosphite hydrate ($Na_2HPO_3 \cdot 5H_2O$; >98 %; Riedel-deHaën 04283, Seelze, Germany) was prepared in deionized water (volume: 20 mL). The pH of the solution was adjusted to 7.0 with sodium hydroxide (NaOH) and nitric acid ($HNO_3$). To initiate the oxidation of phosphite, ca. 572 mg of high-purity iron powder (99+% <200 mesh; Alfa Aesar 00737, Ward Hill, MA) was added into the solution (mg Fe/μmole $HPO_3$ = 0.95). The reaction vessels (Falcon™ 50 mL polypropylene tubes) were tightly sealed to prevent evaporation, that could cause a change in the concentrations of phosphite and the product phosphate. Reaction temperatures were 4 ± 2 °C and 22 ± 2 °C. In order to prevent

the oxidation of phosphite by ambient light, these experiments were performed in the dark.

The concentration of phosphate produced by the oxidation of phosphite was monitored during the course of experiments by extracting a small aliquot of solution (100–150 µL) at 3 days, 35 days and 1901 days (=5 years, 2 months, and 13 days) and performing a colorimetric analysis using the malachite assay[62], at Yale University. We chose this assay because the malachite assay does not affect the oxidation of phosphite during the analyses leading to artifacts, and its analytical precision is ±20 µM (1 SD).

Experiments A9 and A10 (Supplementary Table 2) were designed to determine whether the oxidation of phosphite occurs without iron powder at 4 °C and 22 °C, respectively. Phosphate was not detected over the course of experiment, up to 1901 days (=ca. 5 years and 2 months), suggesting dissolved phosphite is kinetically stable and is not oxidized without iron powder under air up to 1901 days (Supplementary Table 3). For sample A12 (see Supplementary Table 2), iron powder was added in the deionized water to check the purity of iron powder. No phosphate was derived from the iron powder over the course of experiment (Supplementary Table 3).

These results suggest that phosphite is relatively stable even in oxygenated water with half-lives of 1000–3000 years, depending on the temperature, and absent the presence of an oxidizing catalyst capable of forming OH radicals. If the oxidation reaction is first-order with respect to $O_2$, then at $10^{-5} \times$ present atmospheric level[63], these half-lives might be about $10^8$ years, implying very little oxidation over even geologic timescales.

**Biological oxidation model.** We explore the plausibility that phosphite may be oxidized by microbial activity in seawater. Three enzymes have been identified that promote the oxidation of reduced phosphorus compounds: htxD, ptxD, and BAP[17]. Both htxD and BAP use $O_2$ as the electron acceptor in the oxidation process, hence ptxD—which uses $NAD^+$—is the most likely enzyme promoting phosphite oxidation present before GOE[17]. The reaction is effectively:

$$HPO_3^{2-} + NAD^+ == NADH + PO_4^{3-}$$

The rate of oxidation of phosphite is given as[59]:

$$\frac{1}{k} = \frac{202.5}{[HPO_3^{2-}]} + 0.245, \quad (2)$$

where $[HPO_3^{2-}]$ is the initial concentration of phosphite in micromoles/L, and $k$ is the initial oxidation rate or specific activity in micromoles per minute per mg of ptxD, and the concentration of $NAD^+$ is assumed to remain constant at 45 µM[64]. The temperature is assumed to be 35 °C.

The oxidation half-life ($t_{1/2}$) can thus be approximated as:

$$t_{1/2} = \frac{[HPO_3^{2-}]}{2k \times m_{enz}}, \quad (3)$$

where $m_{enz}$ is the mass of the enzyme present per unit volume in the seawater. We find $m_{enz}$ as:

$$m_{enz} = C_d \times m_{cell} \times F_{Pt-ox} \times F_{enz} \quad (4)$$

where $C_d$ is the density of cells in seawater and is assumed[65] to be ~$10^6$ cells/mL, though this is likely a significant overestimate over the depth of the ocean[66]. $m_{cell}$ is the mass of individual bacterial cells, estimated here at about 1 pg, or $10^{-15}$ kg[67]. $F_{Pt-ox}$ is the fraction of cells capable of oxidizing the reduced phosphorus compounds, estimated as 1%[17]. Finally, $F_{enz}$ is the fraction of the cell mass of phosphite-oxidizing microorganisms, that is ptxD. In their enzyme preparation work, Costas et al.[59] retrieved 2–5 mg of ptxD from 20 g of cells by wet weight, though these organisms had overexpressed ptxD formation, and may not be the representative of wild type enzyme production.

Using the above calculations, the biologic oxidation half-life vs. $F_{enz}$ is provided as Supplementary Fig. 5.

**Data availability**. All data generated or analyzed during this study are included within the paper and its Supplementary Information files, and on the corresponding author's Research Gate (https://www.researchgate.net/publication/324017979_Archean_phosphorus_speciation_data).

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

## Acknowledgements

This work was jointly supported by NSF and the NASA Astrobiology Program, under the NSF Center for Chemical Evolution, CHE-1504217, and by National Aeronautics and Space Administration Exobiology grants NNX14AN96G and NNX13AJ36G, and a National Research Foundation of Korea grant (NRF-2015R1D1A1A01060499).

## Author contributions

B.H. and M.A.P. wrote the paper and led the research. S.J.C., R.B., and A.L. procured Archean rock samples. S.J.C. and R.B. determine phosphite oxidation rates in solution. H. A.-L. determined oxidation kinetics of ferrous phosphite. J.S. and Z.A. led HPLC–ICP–MS analysis and Z.A. performed ICP–OES analysis. T.P.K. and B.H. performed NMR and experimental runs. M.A.P. performed modeling.
