## [Peer Review File · Nature Communications]

Reviewers' comments:

Reviewer #1 (Remarks to the Author):

On the basis of experimental data, thermodynamic calculations, and geochemical analyses Herschy and colleagues develop the hypothesis that the cycling of reduced phosphorus (in the form of phosphite) was critical for meeting the nutrient demands of the Earth's early biosphere. They describe a model whereby phosphorus initially bound as ferrous phosphate phases in ferruginous sediments is mobilized at temperatures on the order of ~150-200°C as dissolved phosphite. This phosphite is subsequently transferred to the ocean, and becomes a bioavailable form of P.

The model suggested here is novel, and if true would represent a potentially exciting revision to our understanding of the P cycle on a reducing Earth and our paradigm for the long-term co-evolution between P availability, biospheric productivity, and ocean-atmosphere composition. However, I have significant reservations about each of the major facets that are brought to bear to develop the hypothesis, and unfortunately cannot unreservedly recommend publication unless some of these concerns are dealt with.

First, the authors argue that the presence of phosphite at levels of 1-100ppm dry weight in early Archean supracrustal rocks from Isua and Akilia indicate that phosphite was an important species in the Archean phosphorus cycle. However, it is simply without question that the null hypothesis for the presence of this phosphite should be that it has been produced during burial and metamorphism. Indeed, the authors argue convincingly that Fe(II)-rich rocks containing phosphorus exposed to even relatively mild metamorphic temperatures would be expected to show significant transfer of P from primary phosphate phases to phosphite.

Unless the authors can provide petrographic evidence that the phosphite is a primary phase its presence in what are perhaps the most extremely altered sedimentary rocks in Earth's rock record (indeed, some workers remain unconvinced that the protolith is even sedimentary!) should not be used to suggest anything about primary P chemistry. The authors' argument that "the results here are agnostic of the rocks [sic] origins" is strange, as the protolith and burial/metamorphic history of these units is clearly important for the argument they are making about the presence of phosphite.

My second major concern is that the elevated [P] that the authors posit as sourced from phosphite release seems very reliant on the oxidative half-lives of phosphite in the ocean. I take the authors' point that inorganic oxidation during the Archean was likely limited or negligible, but the rates of biological processing can be quite high. For example, if one were to extrapolate the half-life with respect to biology to the 'abiotic' phosphite concentration in hydrothermal fluids shown in Fig. 3b, the inferred phosphite concentrations would be ~1nM or less for modern high-T water fluxes (and dropping as hydrothermal fluxes increase, which many would consider likely for the early Archean Earth).

There are also some mechanistic details of the model that are somewhat opaque to me. For example, the authors emphasize that their predicted [P] concentrations are well above those posited to exist if P levels are buffered by removal onto iron oxyhydroxides (IOH), their mechanism seems to rely instead on the removal of large amounts of P in association with ferrous phosphate phases. Oxide-facies iron formations (IFs) are notoriously lean in organic carbon, so what reductant to the authors imagine as being operative during burial diagenesis? If phosphite mobilization is metamorphic rather than diagenetic, how does this constrain the extent to which produced phosphite is recycled back into surface environments?

The quantitative model seems to envision a ferruginous sediment dominated by ferrous-phosphate phases, but if it is an oxide-rich system it seems to me the only effect of way of remobilizing P is to form an oxide-rich sediment in the deep sea and alter this near a ridge axis. But it is never

made clear how the thermodynamics of this system would compare to one in which the initial P host is some ferrous phosphate phase.

Lastly, I feel that the discussion of downstream mobility of phosphite is somewhat lacking. What happens to the phosphite produced in each of the above scenarios? What processes could remove it from solution before it makes its way to the photic zone? Are there processes in the deep ocean other than oxidation that can consume it effectively before it reaches the surface ocean?

In sum, although I am receptive to (indeed, excited by) the notion that there is a critical input flux and dissolved, bioavailable phase of P that has not been considered for the early Earth (and perhaps other Earth-like planets), I feel like the evidence from the rock record presented here should be presumed to be severely compromised without additional petrographic data, and the mechanistic model would benefit substantially from additional elaboration in the main text or supplement. I would thus argue that major revisions are required for the manuscript to be published in Nature Communications.

Reviewer #2 (Remarks to the Author):

Authors: Herschy et al.

A very nice paper with extremely interesting results. I have a few recommendations for improvement of the paper, most of which have to do with structural elements and how this relates to the geology of the reported samples. Indeed, the geology of the paper could use a lot of improvement (that part I give a "C", and the chemistry gets an "A"). Overall, the paper with some minor revisions ought to be suitable eventually for publication as it reports on phosphate-phosphite chemistry of distinct interest to origins of life studies.

Throughout, please be consistent and capitalize "Earth"; the alternative "earth" mostly refers to soil.

Change the text "...extensively active on the early earth" to read "...active and commonplace at that time".

Page 2, top. Here the authors could be more precise and up-to-date on what is understood about the nature and timing of the GOE. Firstly, it happened prior to 2.3 Ga based on the multiple S isotope proxy. Reference 11 from Holland was a review paper with no new data. References 12 and 13 report the status of oxygenation in the late Archean (before 2.5 Ga) and not at 2.3 Ga. The correct references would be Kaufman et al. and Papineau et al. both in 2007 in EPSL. They reported data for different continents (Africa and North America, respectively) for the time around 3.42 Ga associated with the ultimate demise of the MIF Sulfur isotopes; such signals never again return as far as we can tell owing to the fact that seawater sulfate had built up to sufficient concentrations to swamp any such signals. Change.

Middle of page 2, paragraph 3. Again, problems with the geology. Although it is correct that Isua is a supracrustal belt, the Akilia rocks are not. Strictly speaking, these are not a succession because there are only a very few rare instances of when these rocks show a classic volcano-sedimentary package or succession. As originally documented by McGregor in the early- to mid-1970s, the Akilia association is a collection of gneissic (including paragneisses) enclaves that are strongly deformed, not clearly part of the Isua supracrustal belt, and that probably pre-date it. To my knowledge, at least one of the co-authors of this paper (Lepland) has been to these rocks, which means that you ought to have got this right.

Furthermore, it also seems that the authors embrace the view that the Akilia rocks are ca. 3.8 Ga or perhaps older. If so, then this needs to be explicitly stated in the paper, otherwise the readership conflates Isua and Akilia and their "potential metasediments".

How can a rock be a "potential" metasediment? Either these rocks are identified as having had a sedimentary protolith, or you just call them what they are: Paragneisses, or mafic schists with associated quartz-pyroxene rocks.

A geologist would not write something like "...the sedimentary origin of these rocks is not proven" because we are not in the business of proving things, only showing that a particular explanation

best comports with the data at hand. From what I can tell, co-authors Blake and Lepland are geologists, which surprises me that this statement appears in this manuscript. The authors go on to state that "the results here are agnostic of the rocks' [sic] origins".

This is broken logic.

Why analyze the rocks from an Eoarchean terrane unless you thought they would provide a signature of the phenomenon you expected to find? You could have collected a sediment from the East Pacific Rise, which is commonly a siliceous-ferruginous ooze, and made the same test. In the very next sentence you state that "(t)hese finding show that high levels of phosphate were... characteristic of the early oceans". How can you claim that if you think these rocks were not originally sediments? This is a logic problem that needs to be fixed.

Next, you cite some work about the "...high flux of meteorites to the Earth in the early Archean", but do you have a good idea of what that flux actually was? I looked at the original reference, and it is far out of date of contemporary understanding of dynamics.

I think that once these logic problems, and errors in citation and attribution are repaired, that the paper will be acceptable for publication.

Reviewer #3 (Remarks to the Author):

The manuscript by Herschy et al. describes the phenomenon of phosphate reduction to phosphite in the presence of ferrous iron, and the relevance of this reaction to increasing the solubility and hence concentration of phosphorus in the Archean ocean. This has implications for the biosphere at that time, as phosphorus limitation has been implicated due to the sorption of phosphate to Fe(III) (oxyhydr)oxide minerals that were likely forming at that time. This is a novel concept, and the application of the phenomenon is sound. It could well have regulated the marine P cycle in early oceans in the face of solution chemistry and sediments dominated by iron. I do have some minor questions and points that need clarification in the text. These are identified below.

First statement of abstract: “The element phosphorus (P) is central to ecosystem growth and is proposed to be the limiting nutrient for life¹” – this statement needs to be qualified. Are you talking about marine life? Primary producers? When in Earth’s history?

Third line of abstract: change “though” to “although”

First line of second paragraph – can you clarify oxidation state/speciation of iron? Given that you are using metallic, zerovalent iron in some experiments?

“These results were geochemically surprising, as the prevalent oxidizing conditions on Earth should confine P to the pentavalent oxidation state, as P occurs principally as the mineral apatite in rocks²²” It would be useful to address the stability of P oxidation states with regard to thermodynamics – perhaps showing an Eh-pH diagram for P overlaid with the of Fe showing the redox boundaries for the species you are referring to.

“However, given the widespread occurrence of ferrous iron prior to the GOE, it is quite likely that reduction of phosphate by low-temperature oxidation of iron is the source of phosphite within Archean rocks.” Please clarify the form that ferrous iron would be in and the relevance of your experiments to the natural system. For instance, you react a precipitated Fe(II)-PO₄ mineral, as far as I can tell from the experiments. Also, the statement “These experiments confirm that moderate heating of phosphate in simulated sedimentary diagenetic or low temperature hydrothermal conditions²⁶ in the presence of ferrous iron can lead to production of phosphite” needs similar clarification. I mention a reference later for authigenic particles forming from ferruginous water columns. You could also find mineralogical data for Fe-rich marine sediments.

You measure and calculate iron phosphite solubility, but have you considered sorption? This seems to be quite important in phosphate removal from oceans, past or present.

Regarding NMR experiments, what is the detection limit of the method? I am not really familiar with this method, but the methods make it seem that there is 80% phosphate, 5% phosphite, and 15% pyrophosphate. The values in ppm don’t really match to this. Furthermore, pyrophosphate has a negative concentration. Please clarify.

Regarding phosphite metal precipitation experiments, how did you ensure that the Fe²⁺ salt did not oxidize with oxygen? Did you do a mass balance of Fe²⁺ oxidized to phosphate reduced? Does this match?

“Given that there is widespread distribution of phosphite-utilizing genes in the modern microbiome²³, and that microbial strains bearing these genes diverged billions of years ago (methods)”. From the section in the methods it appears this is published work. I am unclear why it appears in the methods if it is not new work.

“Current evidence suggests oceans were ferruginous⁴, and given that iron concentrations likely exceeded the saturation point for ferrous phosphate minerals, amorphous ferrous phosphates were likely authigenic in Archean sediments.” It would be good somewhere to state estimates of the range of water column Fe(II) concentrations in the Archean ocean, which certainly was not static in space or time. Also, evidence that authigenic Fe-phosphates formed can be found in analogue environments such as Lake Pavin (see Cosmidis, J.; Benzerara, K.; Morin, G.; Busigny, V.; Lebeau, O.; Jézéquel, D.; Noël, V.; Dublet, G.; Othmane, G., Biomineralization of iron-phosphates in the water column of Lake Pavin (Massif Central, France). *Geochimica et Cosmochimica Acta* **2014**, *126*, (0), 78-96.). Furthermore, reductive dissolution of Fe(III)-oxides in sediments would release both Fe(II) and PO₄³⁻ for further reaction. It would be nice to separate out these two ideas in text in order to better support your assertion that this is a relevant reaction. This paragraph mixes the two ideas.

Methods refers to “Experiments A9 and A10 (Table S1)”, but I believe this is actually ED2?

The phosphite oxidation experiments are confusing. Why utilize Fe₀ powder? As Fe₀ oxidizes, it produces Fe²⁺, which then could back react with the phosphate. It seems it would be good to have an iron-free control experiment. If hydroxyl radicals are the oxidant, there must be other ways to generate them. I would also be interested to know if manganese-oxidizing microbes, which actually produce reactive oxygen species as the chemical oxidant, are also capable of oxidizing phosphite? Actually, there are many, many microbes capable of generating ROS in marine settings (Diaz, J. M.; Hansel, C. M.; Voelker, B. M.; Mendes, C. M.; Andeer, P. F.; Zhang, T., Widespread Production of Extracellular Superoxide by Heterotrophic Bacteria. *Science* **2013**.). Such processes may have driven active phosphite oxidation.

Please find attached our response to the reviews. Reviewer comments are highlighted (yellow), and our response is noted by «>>». Thank you.

Reviewer #1 (Remarks to the Author):

On the basis of experimental data, thermodynamic calculations, and geochemical analyses Herschy and colleagues develop the hypothesis that the cycling of reduced phosphorus (in the form of phosphite) was critical for meeting the nutrient demands of the Earth's early biosphere. They describe a model whereby phosphorus initially bound as ferrous phosphate phases in ferruginous sediments is mobilized at temperatures on the order of ~150-200°C as dissolved phosphite. This phosphite is subsequently transferred to the ocean, and becomes a bioavailable form of P.

The model suggested here is novel, and if true would represent a potentially exciting revision to our understanding of the P cycle on a reducing Earth and our paradigm for the long-term co-evolution between P availability, biospheric productivity, and ocean-atmosphere composition. However, I have significant reservations about each of the major facets that are brought to bear to develop the hypothesis, and unfortunately cannot unreservedly recommend publication unless some of these concerns are dealt with.

First, the authors argue that the presence of phosphite at levels of 1-100ppm dry weight in early Archean supracrustal rocks from Isua and Akilia indicate that phosphite was an important species in the Archean phosphorus cycle. However, it is simply without question that the null hypothesis for the presence of this phosphite should be that it has been produced during burial and metamorphism. Indeed, the authors argue convincingly that Fe(II)-rich rocks containing phosphorus exposed to even relatively mild metamorphic temperatures would be expected to show significant transfer of P from primary phosphate phases to phosphite.

>>This argument is certainly accurate. Our experiments do not go past 200°C, though Herschy (2013) argues that above these temperatures, dimerization of phosphate (forming pyrophosphate) competes with reduction. However, it is likely that metamorphic conditions could result in reduction at higher pressures and temperatures (vs. dimerization, as the formation of water as H₂O(g) during condensation should be less favorable), as our thermodynamic calculations do show (see suppl. Fig 2). In such a case, our findings should still hold- even at relatively low temperatures, reduction occurs readily, and our models predict reduction continuing to higher temperatures. We argue in part that since this reduction is found in Archean rocks, its presence was likely quite common on the early earth. It is likely still produced today, but between an oxidizing environment and biologic activity, it should not last for as long.

Unless the authors can provide petrographic evidence that the phosphite is a primary phase its presence in what are perhaps the most extremely altered sedimentary rocks in Earth's rock record (indeed, some workers remain unconvinced that the protolith is even sedimentary!) should not be used to suggest

anything about primary P chemistry. The authors' argument that "the results here are agnostic of the rocks [sic] origins" is strange, as the protolith and burial/metamorphic history of these units is clearly important for the argument they are making about the presence of phosphite.

>> This is a fantastic idea, but at present the technology to do so convincingly does not exist. Prior work has attempted to investigate intermediate reduced P species that might occur in rocks, but has not demonstrated conclusively that these species are present. Notably, Pasek and Block (Nat. Geosci. 2009) searched for and found phosphites in fulgurites—glasses formed by cloud-to-ground lightning—in occurrences from the American Southwest by bulk rock extracts analyzed by NMR. However, any possible phosphite-bearing grains were too small to measure elemental abundance by microprobe at the levels accurate enough to identify CaHPO_3 vs. CaHPO_4 . Additionally, investigations of meteorite mineral corrosion products (Pirim et al. GCA 2014) did show that the meteoritic mineral schreibersite— $(\text{Fe,Ni})_3\text{P}$ —corrodes to a P-bearing rust. In this case, the O content of the oxide was measured, but, again, couldn't be conclusively identified as the difference between $\text{CaHPO}_3 \cdot \text{H}_2\text{O}$ and CaHPO_4 is little to none by EMPA. We have also sought different P oxidation states out by XPS and Raman, but again, results are inconclusive. Given that the estimated quantity of reduced P is $\sim 1\%$ of the total amount of phosphate, this is a challenge that merits further investigation, but might be beyond the scope of this paper. We do demonstrate by NMR and by HPLC-ICPMS that P is reduced within experiments and rocks, respectively, and do intend (and have made initial progress) to pursue the petrology further.

>> We have addressed/fixed the «agnostic» comment, as well, as per the suggestions of reviewer #2.

My second major concern is that the elevated [P] that the authors posit as sourced from phosphite release seems very reliant on the oxidative half-lives of phosphite in the ocean. I take the authors' point that inorganic oxidation during the Archean was likely limited or negligible, but the rates of biological processing can be quite high. For example, if one were to extrapolate the half-life with respect to biology to the 'abiotic' phosphite concentration in hydrothermal fluids shown in Fig. 3b, the inferred phosphite concentrations would be $\sim 1\text{nM}$ or less for modern high-T water fluxes (and dropping as hydrothermal fluxes increase, which many would consider likely for the early Archean Earth).

>> This is a valid point. The rate of «biological» oxidation of phosphite depends on the microbe doing the oxidation. For instance, White and Metcalf (2007) present a review of the enzymes of phosphite and hypophosphite oxidation by microbes, and provide analysis of three of these enzymes (htxD, PtxD, and BAP). Two of these (htxD and BAP) require O_2 as an electron acceptor, hence they can probably be considered less likely on the early earth environment. The other (PtxD) reduced NAD^+ to NADH with a concomitant oxidation of phosphite to phosphate. There's some evidence that this enzyme is ancient and hence plausibly present on the early earth (Wilson and Metcalf 2005).

We have added a section to our methods on calculating the biological oxidation timescale, finding that bacterial oxidation time scales (half-lives) may be ~ 30 years, but only under optimized conditions (high production of oxidizing enzymes, high cell count over the ocean depth). Otherwise, the oxidation rate is not terribly different from the abiotic oxidation rate. Naturally, there are a number of assumptions in these calculations, but the text has been added to the supporting methods, allowing critique as needed.

Note that our prior diagram has been amended to change the position of the dot from 1 year to 30 years, on account of this more rigorous estimate of peak oxidation rate (we previously used agricultural estimates of phosphite as a fertilizer).

The main text has been modified with the following line: *Biologic oxidation is driven by a few enzymes, some of which require O_2 for redox²³. Biologic oxidation may have been important in the Archean, although calculations of enzymatic oxidation rates suggest a rate of oxidation close to the predicted abiotic oxidation rate (see Supplementary Fig. 6).*

There are also some mechanistic details of the model that are somewhat opaque to me. For example, the authors emphasize that their predicted [P] concentrations are well above those posited to exist if P levels are buffered by removal onto iron oxyhydroxides (IOH), their mechanism seems to rely instead on the removal of large amounts of P in association with ferrous phosphate phases. Oxide-facies iron formations (IFs) are notoriously lean in organic carbon, so what reductant to the authors imagine as being operative during burial diagenesis?

>> The reductant here is ferrous iron, as carbon appears to be inefficient for reduction. So long as phosphite is not present in the initial precipitate, then reduction of phosphate to phosphite should occur (via Lavoisier's principle, leading from disequilibrium chemistry). The Fe^{3+} to Fe^{2+} ratio we used in this model was 1:1, following assumptions from Bjerrum and Canfield, and the initial P^{3+}/P^{5+} ratio was assumed to be zero.

If phosphite mobilization is metamorphic rather than diagenetic, how does this constrain the extent to which produced phosphite is recycled back into surface environments?

>> Phosphite formed through metamorphism must be dissolved in order to be recycled. If metamorphic rocks are subjected to subsequent water-rock interactions, then the phosphite may be removed. We favor diagenesis for this reason, due to the large scale release of water from sediment and low temperatures required for initiating reduction. If the phosphite is formed metamorphically instead of diagenetically, then the extent of water dissolving phosphite should be the driving factor.

The quantitative model seems to envision a ferruginous sediment dominated by ferrous-phosphate phases, but if it is an oxide-rich system it seems to me the only effect of way of remobilizing P is to form an oxide-rich sediment in the deep sea and alter this near a ridge axis. But it is never made clear how the thermodynamics of this system would compare to one in which the initial P host is some ferrous phosphate phase.

>> Our model does assume that the system is effectively an iron oxide and SiO_2 with a few fractions of a percent phosphorus (0.3 % P). We assume then that the phosphite is extracted preferentially (as per our measured solubilities), buffered by the concentration of Fe^{2+} and Fe^{3+} expected from dissolution of oxide minerals.

Lastly, I feel that the discussion of downstream mobility of phosphite is somewhat lacking. What happens to the phosphite produced in each of the above scenarios? What processes could remove it

from solution before it makes its way to the photic zone? Are there processes in the deep ocean other than oxidation that can consume it effectively before it reaches the surface ocean?

>> We do analyze (and produce data on) the possibility of precipitation of phosphite salts (e.g., CaHPO_3 , FeHPO_3 , $\text{Fe}_2(\text{HPO}_3)_3$, and MgHPO_3). Precipitation of phosphite salts/minerals is considered in the models we present, but under no conditions do we predict these salts to form at the sub mM phosphite concentrations we predict. The first phosphite salts to form would be CaHPO_3 or FeHPO_3 , depending on Fe^{2+} concentration, and MgHPO_3 and $\text{Fe}_2(\text{HPO}_3)_3$ aren't predicted to form, as the Mg-phosphite salt is quite soluble, and the Fe^{3+} phosphite salt is buffered instead by formation of Fe(III) oxyhydroxides.

>> The other option is adsorption to oxides/hydroxides (iron or manganese). Abediain et al. (2017) and Barge (pers. comm.) showed that phosphite adsorbs onto iron oxides best/only in the presence of amino acids, in comparison to phosphate, which is more easily removed from solution by IOH adsorption. But more work is needed to better elucidate this process.

In sum, although I am receptive to (indeed, excited by) the notion that there is a critical input flux and dissolved, bioavailable phase of P that has not been considered for the early Earth (and perhaps other Earth-like planets), I feel like the evidence from the rock record presented here should be presumed to be severely compromised without additional petrographic data, and the mechanistic model would benefit substantially from additional elaboration in the main text or supplement. I would thus argue that major revisions are required for the manuscript to be published in Nature Communications.

Reviewer #2 (Remarks to the Author):

Authors: Herschy et al.

A very nice paper with extremely interesting results. I have a few recommendations for improvement of the paper, most of which have to do with structural elements and how this relates to the geology of the reported samples. Indeed, the geology of the paper could use a lot of improvement (that part I give a "C", and the chemistry gets an "A"). Overall, the paper with some minor revisions ought to be suitable eventually for publication as it reports on phosphate-phosphite chemistry of distinct interest to origins of life studies.

Throughout, please be consistent and capitalize "Earth"; the alternative "earth" mostly refers to soil. Change the text "...extensively active on the early earth" to read "...active and commonplace at that time".

>>Capitalized throughout

Page 2, top. Here the authors could be more precise and up-to-date on what is understood about the nature and timing of the GOE. Firstly, it happened prior to 2.3 Ga based on the multiple S isotope proxy. Reference 11 from Holland was a review paper with no new data. References 12 and 13 report the status of oxygenation in the late Archean (before 2.5 Ga) and not at 2.3 Ga. The correct references would be Kaufman et al. and Papineau et al. both in 2007 in EPSL. They reported data for different continents (Africa and North America, respectively) for the time around 3.42 Ga associated with the ultimate demise of the MIF Sulfur isotopes; such signals never again return as far as we can tell owing to the fact that seawater sulfate had built up to sufficient concentrations to swamp any such signals.

>> Thank you. We have changed reference 11 to Papineau et al. 2007., and have modified the text to «*prior to the oxidation of the atmosphere known as Great Oxygenation Event (GOE) around or before 2.4 Ga¹¹⁻¹³.*» We do this in part to tie us to the literature (2.4 Ga), and include the idea of the "whiff" as phosphite could be quite sensitive to oxygenation (thermodynamics suggests as much, though our rate measurements suggest otherwise), and a "whiff" might have been sufficient to alter its chemistry.

Middle of page 2, paragraph 3. Again, problems with the geology. Although it is correct that Isua is a supracrustal belt, the Akilia rocks are not. Strictly speaking, these are not a succession because there are only a very few rare instances of when these rocks show a classic volcano-sedimentary package or succession. As originally documented by McGregor in the early- to mid-1970s, the Akilia association is a collection of gneissic (including paragneisses) enclaves that are strongly deformed, not clearly part of the Isua supracrustal belt, and that probably pre-date it. To my knowledge, at least one of the co-authors of this paper (Lepland) has been to these rocks, which means that you ought to have got this right.

Furthermore, it also seems that the authors embrace the view that the Akilia rocks are ca. 3.8 Ga or perhaps older. If so, then this needs to be explicitly stated in the paper, otherwise the readership conflates Isua and Akilia and their “potential metasediments”.

>> This has been changed, as highlighted below.

How can a rock be a “potential” metasediment? Either these rocks are identified as having had a sedimentary protolith, or you just call them what they are: Paragneisses, or mafic schists with associated quartz-pyroxene rocks.

>>Removed and changed.

A geologist would not write something like “..the sedimentary origin of these rocks is not proven” because we are not in the business of proving things, only showing that a particular explanation best comports with the data at hand. From what I can tell, co-authors Blake and Lepland are geologists, which surprises me that this statement appears in this manuscript. The authors go on to state that “the results here are agnostic of the rocks’ [sic] origins”.

This is broken logic.

Why analyze the rocks from an Eoarchean terrane unless you thought they would provide a signature of the phenomenon you expected to find? You could have collected a sediment from the East Pacific Rise, which is commonly a siliceous-ferruginous ooze, and made the same test. In the very next sentence you state that “(t)hese finding show that high levels of phosphate were... characteristic of the early oceans”. How can you claim that if you think these rocks were not originally sediments? This is a logic problem that needs to be fixed.

>> We have addressed these geology, chronology, and protolith issues, and have expanded the text on geologic descriptions and samples. Specifically, this paragraph has been altered to: “*Here we expand on the suite of early Archean rocks and report data on samples collected from the supracrustal successions of the Isua belt and the Akilia enclave, western Greenland. The age on the Isua Supracrustal Belt is 3.7-3.8 Ga (Nutman et al., 2007) whereas the age of the Akilia enclave is controversial, either c. 3.85 Ga (Nutman et al., 1997, Manning et al., 2006) or c. 3.65 Ga (Whitehouse et al., 1999; 2009). Both Isua and Akilia rocks have experienced deformation, metasomatism and metamorphism that in Isua has reached amphibolites facies and in Akilia granulite facies (Griffin et al., 1980; Rosing et al., 1996; Nutman et al., 1997; Myers, 2001; Fedo and Whitehouse, 2002; Lepland and Whitehouse, 2011). Isua samples were collected from the “low strain” domain (Appel et al., 1998) in the northwestern part of the belt and represent chemical sediments such as BIF and metacherts, and metacarbonates that have metasomatic origin. Apatite occurring in BIFs and metacherts have trace element signatures consistent with the Archean seawater (Lepland et al., 2002). The studied Akilia sample is from the quartz-amphibole-pyroxene unit. The protolith of that unit, either chemical sediment or metasomatised ultramafic rock (Nutman et al., 1997; Fedo and Whitehouse, 2002) has been debated in the literature. The trace element characteristics of whole rock and apatite in best preserved parts of the quartz-amphibole-pyroxene unit are consistent with the sedimentary origin (Friend et al., 2008).*”

Note that we have removed the references in parentheses as per Nature Communication's format, but have left them in above for ease of use.

Next, you cite some work about the "...high flux of meteorites to the Earth in the early Archean", but do you have a good idea of what that flux actually was? I looked at the original reference, and it is far out of date of contemporary understanding of dynamics.

>>This reference has been changed to Pasek 2017, which uses the dynamical model of Marchi et al. (2014) to estimate meteoritic flux. The flux estimated by Marchi et al. (2014) is close to 10% of one crust's worth of meteoritic material since ~4.4 Ga as part of a late-accretionary bombardment. This sentence has been modified to «*in the early Archean and Hadean*¹⁴". It is true that the heavy bombardment at 3.8 Ga, as known a decade ago, has fallen out of favor.

I think that once these logic problems, and errors in citation and attribution are repaired, that the paper will be acceptable for publication.

Reviewer 3.

The manuscript by Herschy et al. describes the phenomenon of phosphate reduction to phosphite in the presence of ferrous iron, and the relevance of this reaction to increasing the solubility and hence concentration of phosphorus in the Archean ocean. This has implications for the biosphere at that time, as phosphorus limitation has been implicated due to the sorption of phosphate to Fe(III) (oxyhydr)oxide minerals that were likely forming at that time. This is a novel concept, and the application of the phenomenon is sound. It could well have regulated the marine P cycle in early oceans in the face of solution chemistry and sediments dominated by iron. I do have some minor questions and points that need clarification in the text. These are identified below.

First statement of abstract: "The element phosphorus (P) is central to ecosystem growth and is proposed to be the limiting nutrient for life¹" – this statement needs to be qualified. Are you talking about marine life? Primary producers? When in Earth's history?

>> This is a good point, and to address this, we've changed the "the" to "a" in "*is proposed to be a limiting nutrient for life¹*", which should encompass some of the variations in "limiting nutrient".

Third line of abstract: change "though" to "although"

>>Changed

First line of second paragraph – can you clarify oxidation state/speciation of iron? Given that you are using metallic, zerovalent iron in some experiments?

>>This has been modified to "*In the nominal reaction, phosphate (HPO_4^{2-}) is reduced to phosphite (HPO_3^{2-}) by the concurrent oxidation of iron (II).*"

"These results were geochemically surprising, as the prevalent oxidizing conditions on Earth should confine P to the pentavalent oxidation state, as P occurs principally as the mineral apatite in rocks²²" It would be useful to address the stability of P oxidation states with regard to thermodynamics – perhaps showing an Eh-pH diagram for P overlaid with the of Fe showing the redox boundaries for the species you are referring to.

>> An Eh-pH diagram is added to extended data (Fig. ED5)

"However, given the widespread occurrence of ferrous iron prior to the GOE, it is quite likely that reduction of phosphate by low-temperature oxidation of iron is the source of phosphite within Archean rocks." Please clarify the form that ferrous iron would be in and the relevance of your experiments to the natural system. For instance, you react a precipitated Fe(II)-PO₄ mineral, as far as I can tell from the experiments. Also, the statement "These experiments confirm that moderate heating of phosphate in simulated sedimentary diagenetic or low temperature hydrothermal conditions²⁶ in the presence of ferrous iron can lead to production of phosphite" needs similar clarification. I mention a reference later for authigenic particles forming from ferruginous water columns. You could also find mineralogical data for Fe-rich marine sediments.

You measure and calculate iron phosphite solubility, but have you considered sorption? This seems to be quite important in phosphate removal from oceans, past or present.

>> Adsorption is certainly possible, but thus far hasn't been investigated by any other researchers. Abediain et al. (2017) and Barge (pers. comm.) showed that phosphite adsorbs onto iron oxides best/only in the presence of amino acids, in comparison to phosphate, which is more easily removed from solution by IOH adsorption. But more work is needed to better elucidate this process.

Regarding NMR experiments, what is the detection limit of the method? I am not really familiar with this method, but the methods make it seem that there is 80% phosphate, 5% phosphite, and 15% pyrophosphate. The values in ppm don't really match to this. Furthermore, pyrophosphate has a negative concentration. Please clarify.

>> In this case, NMR is referenced to a radio frequency spectrum, with H_3PO_4 being set to 0 ppm (or about 161.9 MHz in the magnet used here). Deviations from this are on the order of ppm (hundreds of Hz), determined by shifts in the radio frequency. So the ppm of NMR is useful for identifying chemical speciation, but doesn't detail concentration. Concentration can be determined by integrating beneath peaks on an NMR spectrum. The figure caption has been modified to, "All compounds are references to an external standard of 85% H_3PO_4 (0 ppm, as a frequency spectrum referenced to H_3PO_4 at 161.9 MHz)."

Regarding phosphite metal precipitation experiments, how did you ensure that the Fe^{2+} salt did not oxidize with oxygen?

>> After drying the samples were stored in a glovebox under N_2 and analyzed quickly (2-3 days). After analysis the samples were stored with the others. At this point the samples visibly reddened over the course of about one month, but were blue at the time of dissolution. This has been added to the methods.

Did you do a mass balance of Fe^{2+} oxidized to phosphate reduced?

>> For the reduction experiments, no (these experiments were solids after heating). For the precipitation experiments, yes- the phosphite to Fe^{2+} ratio was 1:1 in this salt, and the ferric phosphite salt was 2:3 (Fe^{3+} to phosphite)

Does this match?

>> These do match the predicted stoichiometries (for precipitation).

"Given that there is widespread distribution of phosphite-utilizing genes in the modern microbiome²³, and that microbial strains bearing these genes diverged billions of years ago (methods)". From the section in the methods it appears this is published work. I am unclear why it appears in the methods if it is not new work.

>> In the case of this data, we applied the “time tree” program database to two disparate organisms. The program is freely available, but we specifically used it to determine the timing of divergence of *P. stutzeri* and *Prochlorococcus*, which, while not difficult, we did not find any references to this approach previously. The cites are to a paper that describes the program (as requested by the authors of the app). Note that there are other suggestions that these genes are ancient, from White and Metcalf (2007), among others cited by this paper. We’ve also added more text to this effect regarding biological oxidation, as requested by reviewer 1.

“Current evidence suggests oceans were ferruginous⁴, and given that iron concentrations likely exceeded the saturation point for ferrous phosphate minerals, amorphous ferrous phosphates were likely authigenic in Archean sediments.” It would be good somewhere to state estimates of the range of water column Fe(II) concentrations in the Archean ocean, which certainly was not static in space or time. Also, evidence that authigenic Fe-phosphates formed can be found in analogue environments such as Lake Pavin (see Cosmidis, J.; Benzerara, K.; Morin, G.; Busigny, V.; Lebeau, O.; Jézéquel, D.; Noël, V.; Dublet, G.; Othmane, G., *Biom mineralization of iron-phosphates in the water column of Lake Pavin (Massif Central, France)*. *Geochimica et Cosmochimica Acta* **2014**, *126*, (0), 78-96.). Furthermore, reductive dissolution of Fe(III)-oxides in sediments would release both Fe(II) and PO₄³⁻ for further reaction. It would be nice to separate out these two ideas in text in order to better support your assertion that this is a relevant reaction. This paragraph mixes the two ideas.

>> This sentence has been modified from the original, but we have added the Lake Pavin reference to the document.

Methods refers to “Experiments A9 and A10 (Table S1)”, but I believe this is actually ED2?

>>- Corrected and changed to ED2

The phosphite oxidation experiments are confusing. Why utilize Fe⁰ powder? As Fe⁰ oxidizes, it produces Fe²⁺, which then could back react with the phosphate. It seems it would be good to have an iron-free control experiment.

>> Note that experiments A9 and A10 were both iron-free. We have found previously that iron (II or III) does not affect phosphite chemistry on the timescales of weeks, but iron metal does (Pasek and Lauretta 2005, Pasek et al. 2007). We posit this to be similar to zero valent iron chemistry wherein iron reacts with O₂ to produce hydroxyl radicals via Fenton-style chemistry.

Note that the iron-free control experiments both had exceedingly little oxidation.

If hydroxyl radicals are the oxidant, there must be other ways to generate them.

>> Prior work involving UV photolysis of water or of gamma irradiation of water also shows phosphite oxidation (e.g., Schwartz and Van der Ween 1972), Schafer and Asmus (1980). So this statement is definitely true.

I would also be interested to know if manganese-oxidizing microbes, which actually produce reactive oxygen species as the chemical oxidant, are also capable of oxidizing phosphite? Actually, there are many, many microbes capable of generating ROS in marine settings (Diaz, J. M.; Hansel, C. M.; Voelker, B. M.; Mendes, C. M.; Andeer, P. F.; Zhang, T., Widespread Production of Extracellular Superoxide by Heterotrophic Bacteria. *Science* **2013**.). Such processes may have driven active phosphite oxidation.

>>This would be a fascinating follow-up with the phosphite chemistry. At present this is unknown. We have added a section detailing phosphite oxidation enzymes in the Archean, but, once the rates of oxidation by potential Mn-generated ROS can be determined, this would be useful for understanding routes of phosphite utilization. As of yet, the phosphite-oxidizing enzymes use O₂ or NAD⁺ as oxidants, but Mn could be quite favorable. Mn oxides do participate in Fenton-style chemistry (Watts et al. *J Env Eng* 2005), which we believe to be a possible major oxidation route (though our case uses Fe).

REVIEWERS' COMMENTS:

Reviewer #1 (Remarks to the Author):

The authors have attempted in good faith to address my concerns and those of the other reviewers. I am happy to see the manuscript published and evaluated by the broader community.

Reviewer #2 (Remarks to the Author):

No further comments. The authors have responded adequately to my review.

Stephen Mojzsis (Boulder, Colorado)

Reviewer #3 (Remarks to the Author):

I am satisfied with the revision and look forward to seeing the article in print.